# Canonical Response Parameterization: Quantifying the structure of responses to single-pulse intracranial electrical brain stimulation

**Kai J. Miller**[1,2]*, **Klaus-Robert Müller**[3,4,5,6], **Gabriela Ojeda Valencia**[2], **Harvey Huang**[7], **Nicholas M. Gregg**[8], **Gregory A. Worrell**[2,8], **Dora Hermes**[2]

**1** Dept of Neurological Surgery, Mayo Clinic, Rochester, Minnesota, United States of America, **2** Dept of Biomedical Engineering & Physiology, Mayo Clinic, Rochester, Minnesota, United States of America, **3** Google Research, Brain Team, Berlin, Germany, **4** Machine Learning Group, Department of Computer Science, Berlin Institute of Technology, Berlin, Germany, **5** Dept of Artificial Intelligence, Korea University, Seoul, Republic of Korea, **6** Max Planck Institute for Informatics, Saarbrücken, Germany, **7** Medical Scientist Training Program, Mayo Clinic, Rochester, Minnesota, United States of America, **8** Dept of Neurology, Mayo Clinic, Rochester, Minnesota, United States of America

\* miller.kai@mayo.edu

**Data Availability Statement:** All code to implement this technique along with the sample data to reproduce the illustrations are publicly

## Abstract

Single-pulse electrical stimulation in the nervous system, often called cortico-cortical evoked potential (CCEP) measurement, is an important technique to understand how brain regions interact with one another. Voltages are measured from implanted electrodes in one brain area while stimulating another with brief current impulses separated by several seconds. Historically, researchers have tried to understand the significance of evoked voltage poly-phasic deflections by visual inspection, but no general-purpose tool has emerged to understand their shapes or describe them mathematically. We describe and illustrate a new technique to parameterize brain stimulation data, where voltage response traces are projected into one another using a semi-normalized dot product. The length of timepoints from stimulation included in the dot product is varied to obtain a temporal profile of structural significance, and the peak of the profile uniquely identifies the duration of the response. Using linear kernel PCA, a canonical response shape is obtained over this duration, and then single-trial traces are parameterized as a projection of this canonical shape with a residual term. Such parameterization allows for dissimilar trace shapes from different brain areas to be directly compared by quantifying cross-projection magnitudes, response duration, canonical shape projection amplitudes, signal-to-noise ratios, explained variance, and statistical significance. Artifactual trials are automatically identified by outliers in sub-distributions of cross-projection magnitude, and rejected. This technique, which we call "Canonical Response Parameterization" (CRP) dramatically simplifies the study of CCEP shapes, and may also be applied in a wide range of other settings involving event-triggered data.

available for use without restriction (other than attribution) at: https://osf.io/tx3yq and https://github.com/kaijmiller/crp_scripts.

**Funding:** KJM was supported by the Van Wagenen Fellowship, the Brain Research Foundation with a Fay/Frank Seed Grant, the Brain & Behavior Research Foundation with a NARSAD Young Investigator Grant, and the Foundation For OCD Research. KRM was supported in part by the Institute of Information & Communications Technology Planning & Evaluation (IITP) grant funded by the Korea Government: No. 2019-0-00079, Artificial Intelligence Graduate School Program, Korea University; and No. 2022-0-00984, Development of Artificial Intelligence Technology for Personalized Plug-and-Play Explanation and Verification of Explanation. KRM was also supported by the German Ministry for Education and Research (BMBF) under Grants 01IS14013A-E, 01GQ1115, 01GQ0850, 01IS18025A, 031L0207D, 01IS18037A as well as Berlin Institute for the Foundations of Learning and Data (BIFOLD). DH & KJM were supported by the Mayo Clinic Center for Biomedical Discovery with a DERIVE grant. This work was also supported by the National Institutes of Health (NIH) NCATS CTSA KL2 TR002379 (KJM), NINDS U01-NS128612 (KJM, GAW), NIMH CRCNS R01MH122258 (DH), NIH UH2/UH3-NS95495 (GAW), and R01-NS09288203 (GAW). Manuscript contents are solely the responsibility of the authors and do not necessarily represent the official views of the NIH. The funders had no role in study design, data collection and analysis, decision to publish, or preparation of the manuscript.

**Competing interests:** The authors have declared that no competing interests exist.

## Author summary

We introduce a new machine learning technique for quantifying the structure of responses to single-pulse intracranial electrical brain stimulation. This approach allows voltage response traces of very different shape to be compared with one another. A tool like this has been needed to replace the status quo, where researchers may understand their data in terms of discovered structure rather than in terms of a pre-assigned, hand-picked, feature. The method compares single-trial responses pairwise to understand if there is a reproducible shape and how long it lasts. When significant structure is identified, the shape underlying it is isolated and each trial is parameterized in terms of this shape. This simple parameterization enables quantification of statistical significance, signal-to-noise ratio, explained variance, and average voltage of the response. Differently-shaped voltage traces from any setting can be compared with any other in a succinct mathematical framework. This versatile tool to quantify single-pulse stimulation data should facilitate a blossoming in the study of brain connectivity using implanted electrodes.

This is a *PLOS Computational Biology* Methods paper.

## Introduction

Electrical stimulation of the brain can be used for a variety of diagnostic, therapeutic, and scientific purposes. Interactions between brain regions may be studied by applying or inducing pulses of electrical stimulation to a particular site, while measuring the electrophysiological response at the same place or elsewhere [1–3]. In particular, the averaging of measured voltages from implanted electrodes following brief (several millisecond) pulses of current produces widespread but sparse deflections from baseline (Fig 1). These voltage traces are typically called "single-pulse electrical stimulation" responses or "cortico-cortical evoked potentials" (CCEPs) [4–6]. We make measurements of these types with recordings of the convexity brain surface electrocorticography (ECoG) or in deeper structures from stereoelectroencephalography (stereoEEG; sEEG) and deep brain stimulation (DBS) electrodes with our neurosurgical patients [7]. Despite the "CCEP" name, these stimulation-evoked potential changes are seen with stimulation and recording of non-cortical structures such as white matter, basal ganglia, thalamus, and others [8, 9]. Contemporary analysis of these CCEP responses has suffered from reliance on pre-defined assumptions about the shape that the response should have and quantification of effect only by the voltage at a particular time. This manuscript describes an algorithmic approach to formalize and simplify CCEP analysis so that responses of different shape, duration, and magnitude may be quantitatively compared with one another.

Because stimulation studies often involve a very large number of stimulated-at and measured-from brain sites, the potential set of interactions to study can become very large and make it difficult to examine data to discover simplifying principles. To address this, we recently introduced a conceptual framework formalizing four basic paradigms for interpreting CCEP data [10]:

- The hypothesis-preselected paradigm—Two brain sites are chosen based upon a pre-defined anatomical or functional hypothesis, and a 1-way or 2-way interaction between them is characterized.

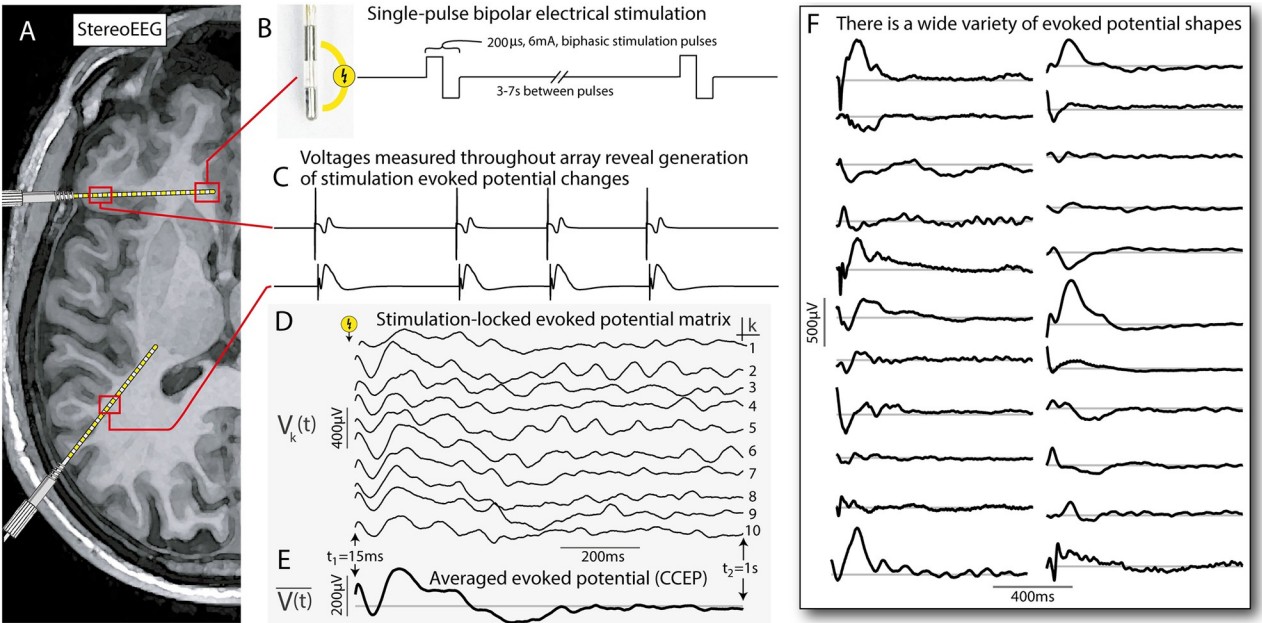

**Fig 1. Single-pulse electrical stimulation with stereoelectroencephalography (sEEG). A**. A cartoon schematic of an axial MRI with two sEEG leads. **B**. Single-pulse biphasic electrical stimulation is delivered through adjacent sEEG electrode contacts (200$\mu$s, 6mA), separated by 3–7s between pulses. **C**. Cartoon voltage traces that might be elicited at two different sites in response to stimulation at a third site (i.e. with a stimulation artifact followed by a characteristic evoked potential deflection). **D**. An example set of actual evoked potentials showing the stimulation-locked evoked potential matrix **V**, with columns $V_k(t)$) shown as individual traces. **E**. Average stimulation-evoked potential from (D). **F**. Examples of some of the different measured average response shapes seen in these studies (as in E). These selected responses were produced from 5 different stimulation sites across two patients (over the interval 15ms-1s post-stimulation, where the gray line indicates 0 $\mu$V). The variety of different shapes seen in just this small subset shows that there is no one typical form of stimulation evoked potential shape.

- The divergent paradigm—Stimulation is performed at one brain site and measured responses at all sites are examined and compared. For $N$ brain sites, this characterizes $N$ interactions.

- The convergent paradigm—One brain site is measured from, and the effects of stimulations at all brain sites are compared versus one another based upon the response shapes at the measured-from site. For $N$ brain sites, this characterizes $N$ interactions.

- The all-to-all paradigm—All brain sites are stimulated at, and responses are measured at all sites. For $N$ brain sites, this characterizes $N^2$ interactions.

We previously addressed the *convergent* paradigm [10], which allows one to uncover "Basis Profile Curves" (BPCs) whose shapes characterize different types of responses at a measured-from brain site that can be intuitively mapped back anatomically to the stimulated brain sites. However, for many studies, what is needed is a simple way to characterize the structure of an evoked response at a single measured-from brain site produced by stimulating at one brain site (the *hypothesis-preselected* paradigm). This manuscript addresses this need with a new technique for identifying structure in an evoked timeseries and parameterizing single trials in terms of it.

Previous quantifications of voltage deflections in single-pulse responses (CCEPs) have typically assumed a single canonical shape consisting of characteristic negative deflections between $\sim$10–100 ms from stimulation called the "N1" response and a later second negative deflection (called the "N2") [4, 11, 12]. However, there are a wide variety of evoked potential

shapes in CCEP responses, and the N1/N2 description is insufficient to describe most of them, as seen in Fig 1. There has not been an alternate, generic, way of approaching these data in the time domain (though some have proposed generic frequency-domain approaches [13]). A formulation is needed that studies a set of repeated trials of stimulation and extracts a canonical structure in the response (if one exists), without a pre-set assumption of the response shape. Our proposed method, which we call "canonical response parameterization" (CRP) provides a recipe for examining structural similarity between trials to a) identify whether there is a significant reproducible response shape (and over what time interval), b) characterize what this shape is, and c) parameterize single trials by the weight of the discovered shape and the residual (after the discovered shape has been regressed out). Equipped with our novel CRP parameterization, researchers can quantify the magnitude, duration, and significance of response to stimulation between pairs of brain sites in a generic framework.

## Materials and methods

### Ethics statement

The study was conducted according to the guidelines of the Declaration of Helsinki, and approved by the Institutional Review Board of the Mayo Clinic IRB# 15–006530, which also authorizes sharing of the data. Each patient / representative voluntarily provided independent written informed consent to participate in this study as specifically described in the IRB review (with the consent form independently approved by the IRB).

### Measurement of cortico-cortical evoked potentials

Two patients with epilepsy (19 and 63 years old, both male) participated in the study while undergoing monitoring to localize their seizure onset zone. Patient 1 was implanted with 15 bilateral sEEG leads and patient 2 was implanted with 13 left sided sEEG leads plus several scalp EEG electrodes (in canonical 10/20 locations). Each sEEG leads consisted of 10–18 contacts, composed of cylindrical platinum-iridium electrodes of 2 mm length, with 1.5 mm between (3.5mm center-to-center separation). The diameter of the lead is 0.8 mm, giving each contact has an exposed surface area of 5.0 $mm^2$ (Fig 1). Electrode recordings were excluded if they were: 1) in seizure onset zone, 2) not stimulated, 3) artifactual, or 4) not in the brain (i.e. not extended past the bolt, etc). Voltage data were recorded at 2048Hz with a Natus Quantum amplifier. Electrode pairs were stimulated 10 times with a single biphasic pulse of 200 microseconds duration and 6 mA amplitude every 3–7 seconds using a Nicolet Cortical Stimulator. Data first were notch filtered to remove 60Hz line noise and then re-referenced to a modified common average on a trial-by-trial manner to exclude stimulated channels and channels with large variance, as described in prior work [14]. Electrodes were localized on post-operative CT scans and coregistered to preoperative MRI using the *sEEG View* package [15], available on github [16]. All code to implement this technique along with the sample data to reproduce the illustrations are publicly available for use without restriction (other than attribution) at: https://osf.io/tx3yq and https://github.com/kaijmiller/crp_scripts.

Our illustration of this technique is limited to data from two patients. However, the reader interested in applying this technique with a wider set of example data than that included here can access further illustrative recordings released with our other emerging work by van Blooijs et. al. [17], Ojeda Valencia et. al. [18] and Huang et. al. [14].

## Data structure

The quantification of interaction between a stimulated brain site and a recorded brain site begins with a matrix of single-trial voltage responses. Matrix $V_k(t)$ is drawn from the voltage data from the measured brain site, selecting epochs of time $t$ over the naïvely chosen interval $t_1$ to $t_2$ following the time $\tau_k$ of the $k^{th}$ stimulation from the stimulated electrode pair, where $t$ denotes the time from the $k^{th}$ electrical stimulation at brain site $m$, $\tau_k$: $(\tau_k + t_1) \leq t \leq (\tau_k + t_2)$. The dimensions of $\mathbf{V}$ are $T \times K$, with $T$ total timepoints (over the interval $t_1 \leq t \leq t_2$) by $K$ total stimulation events (Fig 1D). This fragment is an error. In this manuscript, $t_1$ of 15ms was chosen to reduce the likelihood of contamination by stimulation artifact, and $t_2$ of 1s was chosen because (anecdotally) the vast majority of sEEG CCEPs we have observed return to baseline well before then. We anticipate that researchers will adjust $t_1$ and $t_2$ based upon their own circumstances. Before data are analyzed and parameterized, one should evaluate the preprocessed data for baseline fidelity. Issues with inappropriate referencing or lack of a baseline around zero can be avoided by visual inspection or calculation of the mean and/or median values at far from stimulation times.

## Single-trial cross-projections

In order to understand shared structure between stimulation trials, we first obtain a matrix of unit-normalized single trials: $\tilde{V}_k(t) = V_k(t)/|V_k(t)|$. Each $\tilde{V}_k(t)$ is then projected into all other trials, $\mathbf{P} = \tilde{\mathbf{V}}^\top \mathbf{V}$:

$$P(k,l) = \sum_t \tilde{V}_k(t) V_l(t)$$

Note that $P(k,l) \neq P(l,k)$. The full matrix $\mathbf{P}$ is subsequently sorted into a combined set $S$, with self-projections ($k = l$) omitted, and a total of $K^2 - K$ elements (Fig 2). Each element (initially with units $\mu V \cdot \sqrt{\text{\# samples}}$) is then scaled by $1/\sqrt{\text{samplerate}}$ so that it carries the units $\mu V \cdot \sqrt{s}$. The average over the set of cross-projections, $\overline{S}$, summarizes the interaction from stimulation to response.

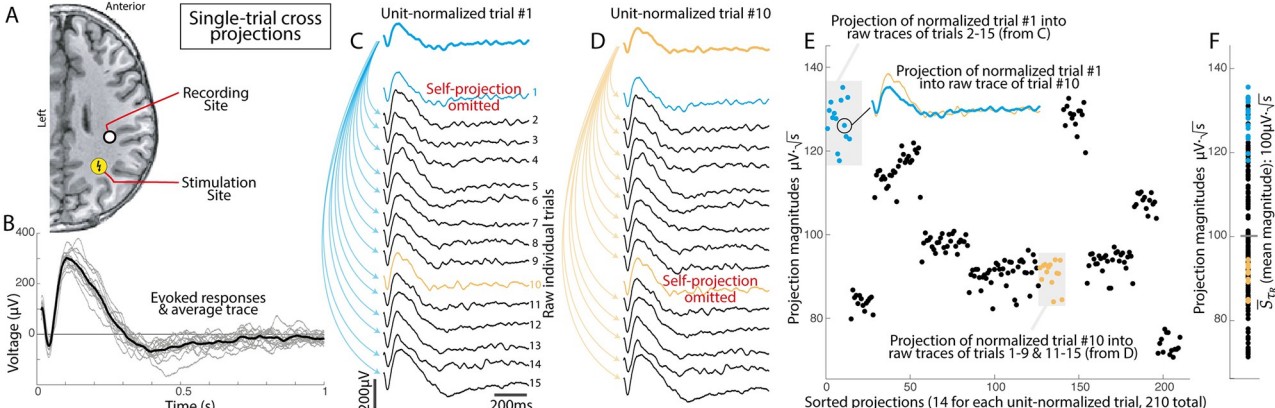

**Fig 2. Quantifying single-trial cross-projections. A**. Stimulation and recording sites for this example, shown in an axial MRI section. **B**. 15 single trials of stimulation response (gray) produced the averaged evoked potential shape (black). **C**. Trial #1 (light blue) was unit-normalized and projected into the other 14 trials, omitting self-projection. **D**. As in (C), but for normalized trial #10 (orange). **E**. All 210 projections are shown sorted, note the obvious sub-sets corresponding to the projections of unit-normalized single trials. The projections of each trial into the others can reflect how representative each trial is of the canonical evoked potential response shape. **F**. The projections from (E), aggregated into a single column (i.e. imposing the assumption that the order of trials doesn't matter, which will be false under some circumstances).

## Response duration

In order to quantify how long there is a significant effect after stimulation, the set $S$ can be constructed over different time periods to determine the duration of most statistically-meaningful response. We do this by determining projection weights $S$ and $\overline{S}$ as a function of time, $S \rightarrow S_t$, and quantify a temporal profile, $\overline{S}_t$, as illustrated in Fig 3. Because $\overline{S}$ may be thought of as a reflection of mutual information between responses (i.e. their correlated deviation from 0V), the peak of $\overline{S}_t$ represents the time past which further information is not reliably contained in the response—when the distribution of voltages across responses drifts to be indistinguishable from $0\mu$V. We define this peak time as the "*response duration*", or $\tau_R$. It is important to note that this CRP approach is very sensitive to baselining, as the sign of the voltage in cross-projections around zero determines when the profile of $\overline{S}_t$ begins to decrease. The uncertainty of $\tau_R$ could be estimated in many ways, but, for the illustrations in this manuscript, we place error bars where $\overline{S}(t_2)$ exceeds 98% of $\overline{S}(\tau_R)$. The truncated voltage matrix representing just the times in the response to stimulation up to this duration $\tau_R$ is henceforth denoted **V**, with dimensions $\tau_R - t_1$ ("$T_R$") timepoints and $K$ trials. The initial voltage matrix (over the naïvely-chosen time interval) will be specifically designated as such when discussed further in the text.

## Extraction significance: Quantifying reliability of response structure and identifying anomalous single trials

The set of projection magnitudes $S_{\tau_R}$ can be tested against zero for significance, which we call the "*extraction significance*". Note that one cannot use the full distribution of $S_{\tau_R}$—this creates

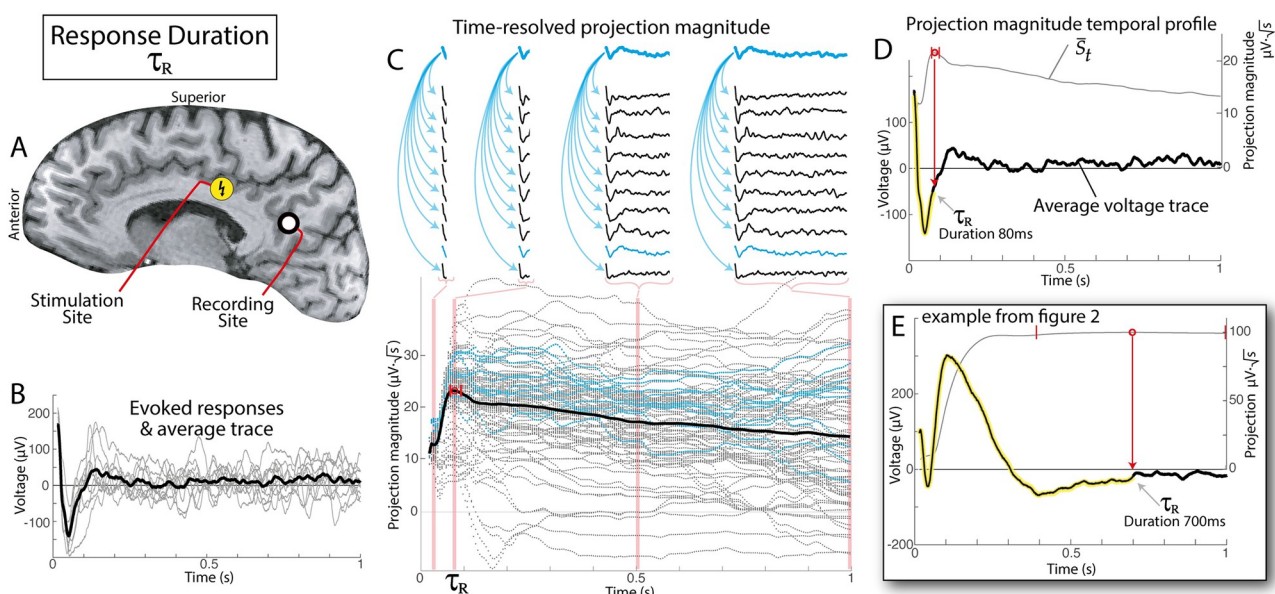

**Fig 3. Using time-resolved projection weight to quantify response duration, $\tau_R$. A.** Stimulation and recording sites for this example, shown in a sagittal MRI section. **B**. 10 single trials of stimulation response (gray) produced the averaged evoked potential shape (black). **C**. Abbreviated timeseries are calculated from $t_1$ to a range of $t_2$s to obtain time-resolved projection weights (individual dots). The traces above indicate a subset of the projections (for the normalized trace of $9^{th}$ trial) at times $t_2 = 20$ms, $t_2 = 80$ms $= \tau_R$, $t_2 = .5$s, and $t_2 = 1$s, with distributions of $S_{t_2}$ at each of these timepoints highlighted in the light red background. In this example, the blue dots are projection of the normalized trace of $9^{th}$ trial (illustrated in traces above). The thick black line is $\overline{S}(t_2)$. Calculated response duration, $\tau_R$, is indicated by a red circle. Small vertical red lines indicate thresholds where $\overline{S}(t_2)$ exceeds 98% of $\overline{S}(\tau_R)$ (providing an estimate of the error in calculating $\tau_R$). Note that blue dots in bottom portion are the projections illustrated for the $9^{th}$ trial from the traces on the top. **D**. The projection weight temporal profile, from the black line in the lower portion of (C), is shown with a gray line. The averaged voltage response, from the black line in (B), is shown with a black line, and the significant portion of the response is highlighted (i.e. up to $\tau_R$). **E**. As in (D), but for the example response from Fig 2.

artificial significance since a pair-wise interaction between two trials is counted twice. Once when the once when the first trial is normalized & the second trial is raw, and once in the inverse case. Therefore, only half of projections are considered for p-value and t-value analysis. These are selected such that 1) each trial is the normalized trial half of the time and raw half of the time, and 2) pair-wise interactions are counted only once. Sub-distributions of $S_{\tau_R}$ corresponding to the projection magnitudes involving single trials can be used to identify the "most anomalous and most normal" single trials—(with comparison against the same selection of half of the projection magnitudes). This can be used as a simple technique for artifact rejection.

## Identification of canonical CCEP shape using Linear Kernel PCA

We would like to identify a characteristic shape of the canonical CCEP, $C(t)$, determined from **V**, that characterizes a stimulation-induced interaction between brain regions. The most common way to do this is to take the simple average trace (i.e. $C(t) \rightarrow \overline{V(t)}$). However, we prefer a quantity that represents the "principal direction" ($1^{st}$ principal component) of **V**, which captures the variance of the data and is more robust than the average against outlier trials. A standard principal component decomposition (PCA, [19]) is generally not possible in these data because of the practical fact that the number of timepoints, $T_R$, generally far exceeds the number of trials, $K$, in these data (i.e. $T_R \gg K$), which would require $K > T_R^2$ to characterize the $T_R$-by-$T_R$ matrix of interdependencies between timepoints in PCA. As in prior work [10], we address this issue by inverting the decomposition using the *Linear Kernel PCA* technique [20–22]. This method allows for the interchange of an eigenvalue decomposition of the matrix $\mathbf{V}\mathbf{V}^\mathsf{T}$ ($T_R^2$ elements) with $\mathbf{V}^\mathsf{T}\mathbf{V}$ ($K^2$ elements). Following this approach, we obtain a matrix **F**, whose columns are the eigenvectors of $\mathbf{V}^\mathsf{T}\mathbf{V}$, with associated eigenvalues contained in the diagonal matrix $\boldsymbol{\xi^2}$, satisfying $(\mathbf{V}^\mathsf{T}\mathbf{V})\mathbf{F} = \mathbf{F}\boldsymbol{\xi^2}$. We can then solve for the eigenvectors of $\mathbf{V}\mathbf{V}^\mathsf{T}$, contained in the columns of **X**: $\mathbf{X}\boldsymbol{\xi} = \mathbf{V}\mathbf{F}^\mathsf{T}$. We keep the first column of **X** as our canonical CCEP shape $C(t)$ (Fig 4).

## Parameterizing single trials in terms of the canonical response shapes

We utilize the formalism from functional data analysis to parameterize our data [23, 24]. Each individual trial is represented as a projection of a canonical CCEP form $C(t)$, scaled by a scalar $\alpha_k$, with residual $\varepsilon(t)$ (note that $\varepsilon(t)$ reflects combined measurement noise and uncorrelated brain activity):

$$V_k(t) = \alpha_k C(t) + \varepsilon_k(t)$$

We assert that the expectation values related to $\varepsilon(t)$ are $E(\varepsilon) = 0$ and $E(\varepsilon_k^2) \sim E(\varepsilon_l^2)$, for all $k$ and $l$. This allows us to estimate the projection of $C(t)$ into each individual trial as follows. First, we expand our single-trial formalism above by application of $\sum_t C(t)$ to both sides, i.e.:

$$\sum_t C(t) V_k(t) = \sum_t C(t)\alpha_k C(t) + \sum_t C(t)\varepsilon_k(t)$$

However, $\sum_t C(t)\varepsilon_k(t) = 0$ since $E(\varepsilon) = 0$, and $\sum_t C(t)\alpha_k C(t) = \alpha_k \sum_t C(t)C(t)$, which is just $\alpha_k$, since $\sum_t C(t)C(t) = 1$. This allows us to calculate $\alpha_k$ for each trial:

$$\alpha_k = \sum_t C(t) V_k(t)$$

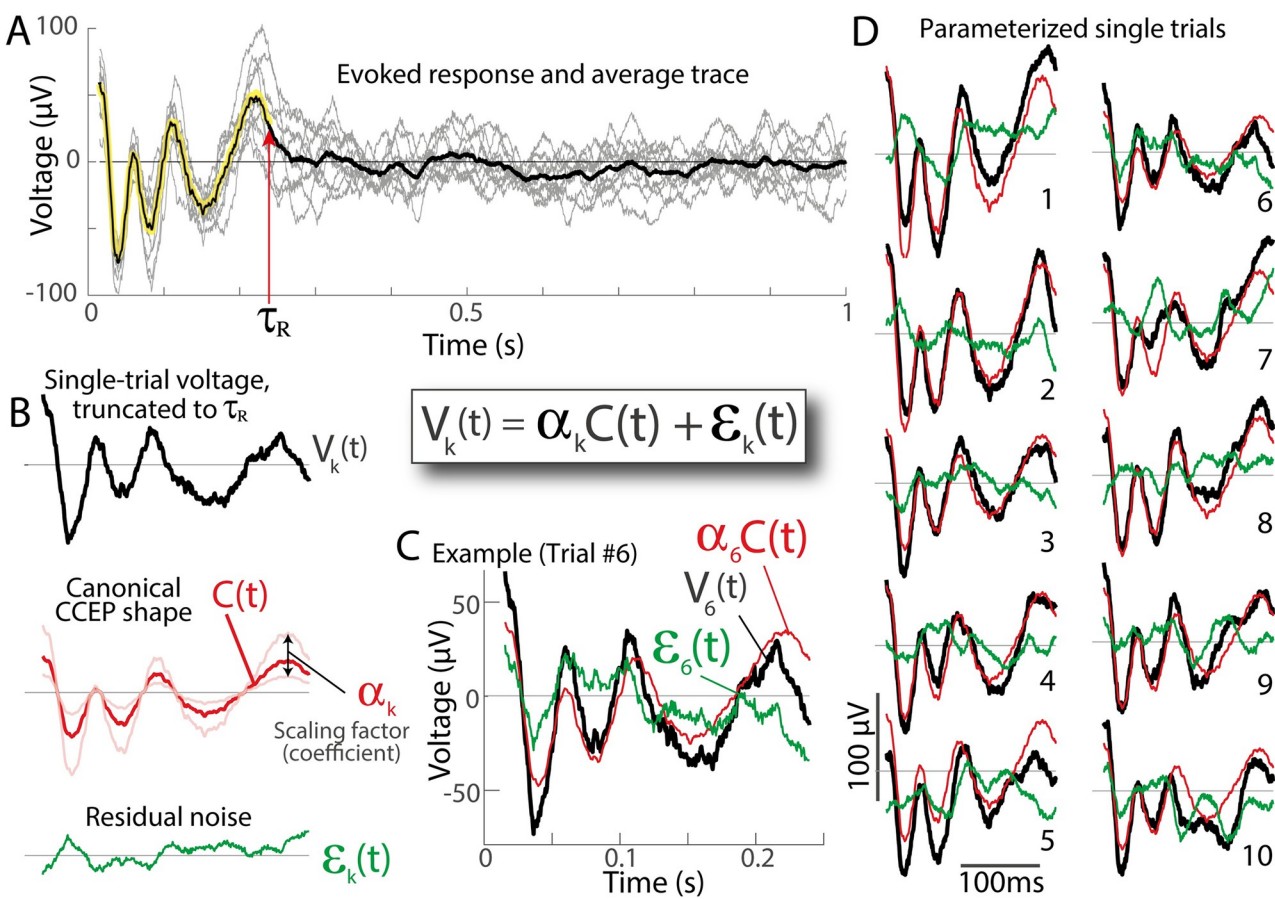

**Fig 4. Parameterizing the evoked response for single trials. A**. An example evoked response, as in Figs 2B and 3B. **B**. The voltage response $V_k(t)$ from trial $k$ (black) is parameterized by how strongly the canonical response shape ($C(t)$, red trace, time interval $t_1$ to $\tau_R$) is represented (scaling factor $\alpha_k$) plus the "residual" $\varepsilon_k(t)$ (green): $V_k(t) = \alpha_k C(t) + \varepsilon_k(t)$. **C**. Overlaid $V_k(t)$, $\alpha_k C(t)$, and $\varepsilon_k(t)$ for example trial #6. **D**. As in (C), for all 10 trials.

Knowing $\alpha_k$, we can quantify the residual signal after regressing out the shape of $C(t)$:

$$\varepsilon_k(t) = V_k(t) - \alpha_k C(t)$$

With the description $V_k(t) = \alpha_k C(t) + \varepsilon_k(t)$, several useful quantities for each trial $V_k$ can be described: a "projection weight" $\alpha_k$; a scaled version of projection weight, $\alpha'_k$ that is normalized by the square root of the number of samples in $C(t)$ (i.e. in $t_1$ to $\tau_R$ interval) and carries intuitive units of $\mu$V (analogous to root-mean-squared response); a scalar "noise" summary term $\sqrt{\varepsilon_k^\mathsf{T} \varepsilon_k}$ (magnitude of the residual); a "signal-to-noise" ratio $\alpha_k / \sqrt{\varepsilon_k^\mathsf{T} \varepsilon_k}$; the "explained variance" by the canonical stimulation response (CCEP shape) is $1 - \frac{\varepsilon_k^\mathsf{T} \varepsilon_k}{V_k^\mathsf{T} V_k}$. Table 1 summarizes these discovered parameters, with some examples illustrated in Fig 5. Note that canonical response extraction and parameter discovery can be highly sensitive to baselining appropriately (Fig 6). The distributions of single trial parameters can be used to quantify the significance of the canonical shape to explain variation in the data, and we call these measures "*parameterization significance*" to distinguish them from the extraction significance that is described above.

**Table 1. Discovered parameters for single stimulation-recording pair.**

| Parameter | Units | Interpretation |
|---|---|---|
| $\tau_R$ | s | Response duration |
| $\overline{S}_{\tau_R}$ | $\mu V \cdot \sqrt{s}$ | Averaged[1] cross-projection magnitude at time $\tau_R$ |
| $C(t)$ | $1/\sqrt{\#\ samples}$ | Canonical CCEP form (unit norm vector, length $\tau_R$) |
| **Single-trial parameters** (for trial k) | | |
| $\alpha_k$ | $\mu V \cdot \sqrt{\#\ samples}$ | Projection weight (how strong $C(t)$ is represented in trial $k$). |
| $\alpha'_k$ | $\mu V$ | $\alpha_k$ normalized by $\sqrt{\#\ samples}$ in $t_1$ to $\tau_R$ interval. |
| $\varepsilon_k(t)$ | $\mu V$ | Residual data after regressing out the shape of $C(t)$. |
| $\sqrt{\varepsilon_k^\intercal \varepsilon_k}$ | $\mu V \cdot \sqrt{\#\ samples}$ | A scalar single trial summary term quantifying the magnitude of the residual. |
| $\alpha_k / \sqrt{\varepsilon_k^\intercal \varepsilon_k}$ | dimensionless | "Signal-to-noise" ratio of projection weight to residual. |
| $1 - \frac{\varepsilon_k^\intercal \varepsilon_k}{v_k^\intercal v_k}$ | dimensionless | "Explained variance" by the stimulation response $C(t)$. |

[1] Averaged over all non-self stimulation trial cross-projections.

## Results and discussion

The figures in this manuscript show a wide variety of shapes in evoked voltage responses to brain stimulation. No single or several pre-defined form(s) would adequately capture the shape of these responses, with or without sign-flips, temporal scaling, or other manipulation. Therefore, we have constructed an approach to extract structure from these data that begins by calculating semi-normalized dot-product projections between single trials over increasing time intervals. From this, we can uncover a duration of significant response $\tau_R$, extract a characteristic shape $C(t)$, and then parameterize the single trials in an intuitive formalism: $V(t) = \alpha C(t) + \varepsilon(t)$ (Fig 4). We call this recipe "Canonical Response Parameterization" (CRP).

### Projection magnitudes and response duration

Description of the voltage time series response to a stimulus typically begins by visualization of the average of many repeated stimuli (Fig 1). In practice, one then tries to infer how robust this shape is by visually observing a suppression of "roughness" in the small deviations in shape as more trials are added to averaging. Alternately, one can plot single trials in the background of the average shape to quantify trial-to-trial variability from the mean (as in Figs 2B or 3B, for example). However, it would be preferable to have a direct quantification of similarities between different trials, and our technique addresses this by performing pairwise cross-projections between trials (with one normalized) to identify structure. The distribution of these cross-projection magnitudes can be compared versus zero to determine significance, which we call *extraction significance* (illustrated in Fig 2). By omitting self-projections, there is no self-consistency in significance determination and no appeal to the mean across all trials.

This projection technique is then further elaborated upon by applying it to limited time epochs for comparison, as illustrated in Fig 3. A temporal profile for projection magnitude results from this and illustrates the accumulation of information as more structure is considered in the comparison. When adding further time includes data where structure is lost as individual traces trend across zero, negative contributions to the dot product produce a decrease in the overall cross-projection magnitude. The timepoint of the maximum of the temporal profile of cross-projection magnitude therefore reveals the end of the time epoch that is meaningful across trials (we call this the "response duration" $\tau_R$, Figs 3 and 4).

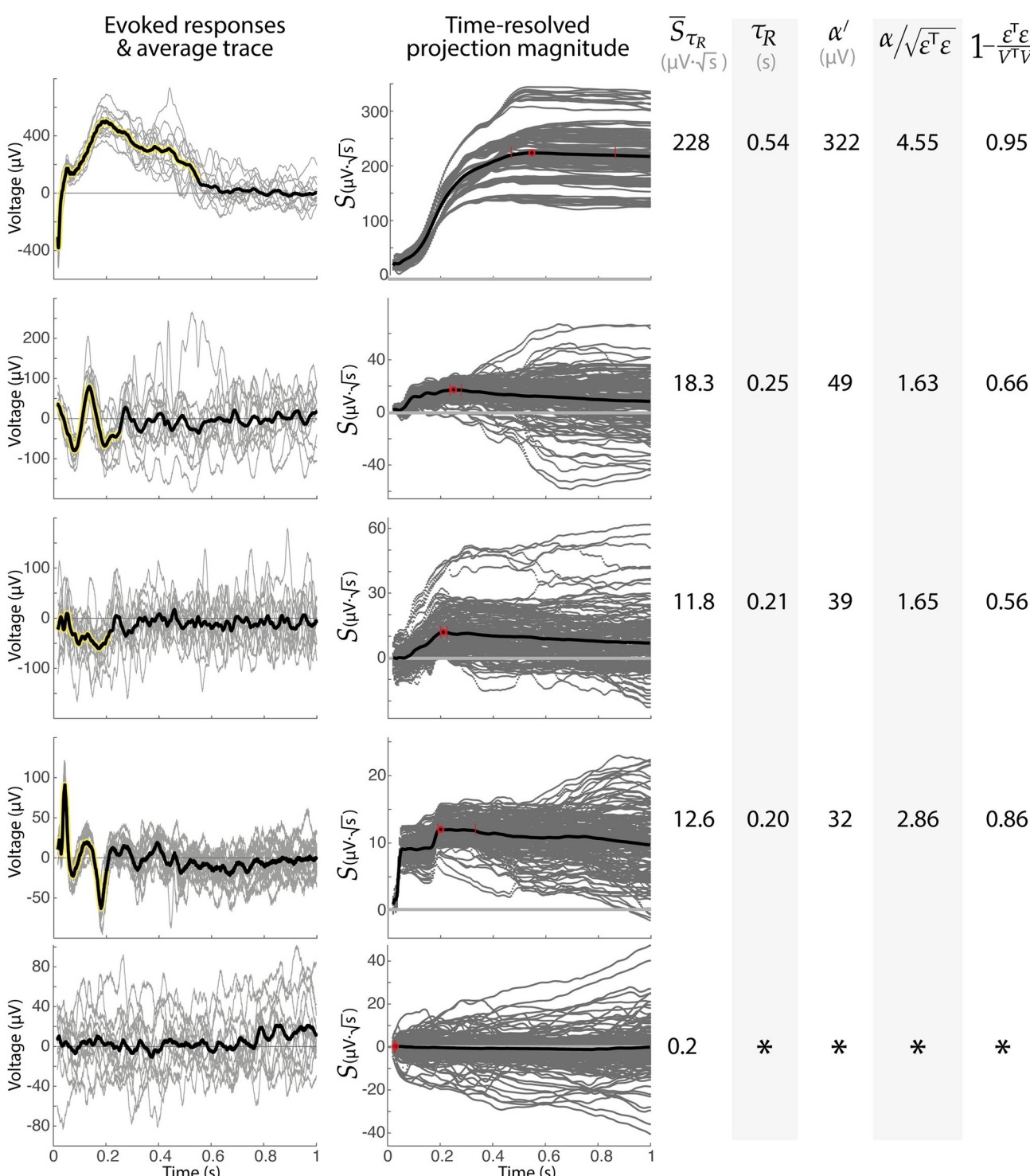

**Fig 5. Examples of shapes, durations, projections, and parameterizations.** Five example responses illustrate projection magnitude profiles, parameterization values, and significance metrics. Note that the bottom response does not meet signficance at any time. The four top examples all met extraction significance at $\tau_R$ of p$\ll 10^{-16}$ (t-test of $\overline{S}(\tau_R)$ vs 0). The bottom example is not significant (p = 0.37). Single trial parameters are averaged across trials for the 3 right-most columns. Note that the second trace might be called the classic N1/N2 response.

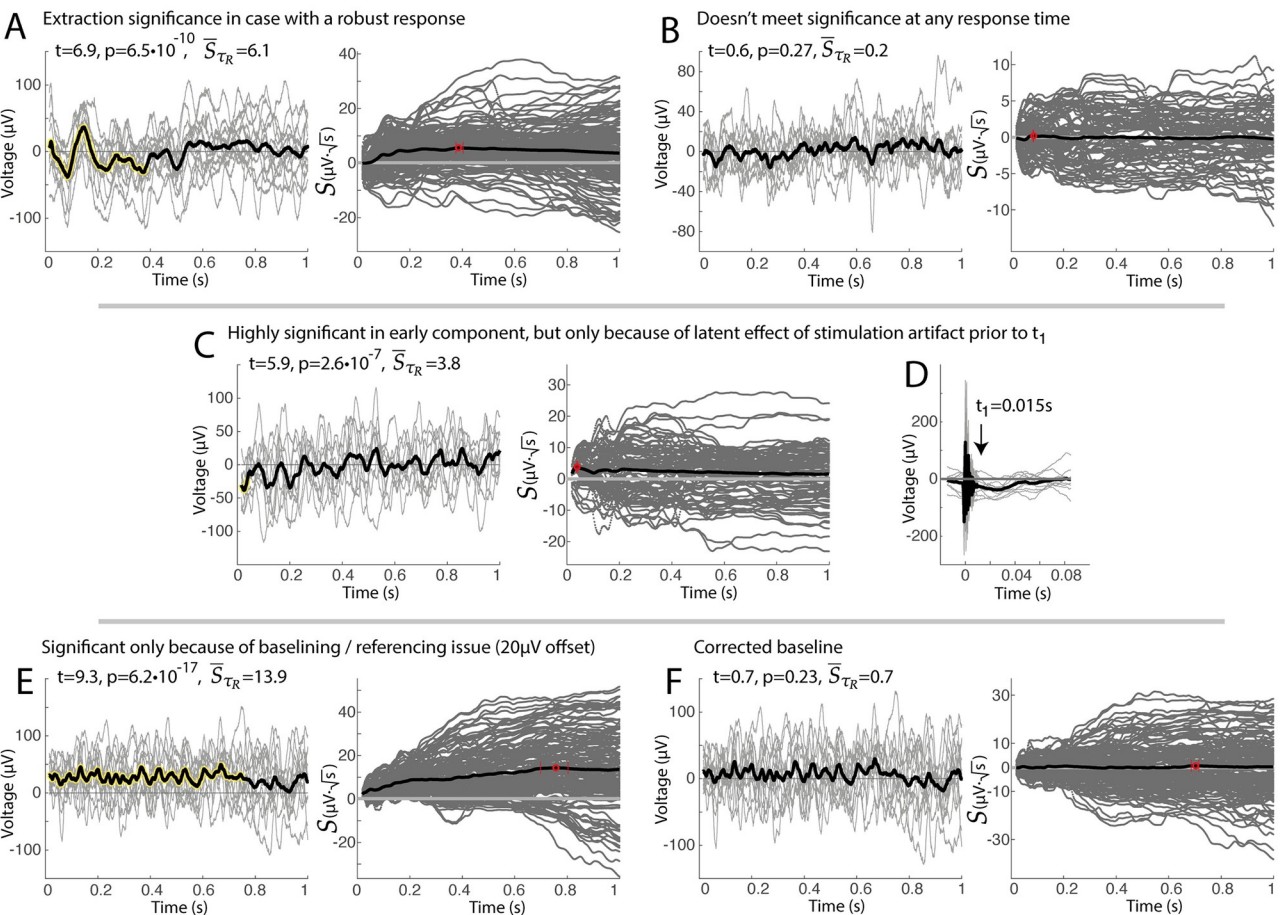

**Fig 6. Illustrative examples of extraction significance. A**. An example of a high noise, but highly significant voltage response. **B**. An example of no significant response to stimulation. **C**. Early significance is detected in an apparently insignificant response. **D**. Examination of the voltages prior to $t_1$ shows a clear (presumably artifactual) offset, explaining the observation in (C). **E**. An example of significance throughout a response that appears to be insignificant, though does have a non-zero offset. **F**. Correcting (E) for the $20\mu V$ offset in baseline removes the artifactual significance. Note that p-values determined by t-test of $\overline{S}(\tau_R)$ vs 0.

Figs 5 and 6 show that significant vs. insignificant trials can be readily identified by applying simple statistics to the cross-projection magnitudes, $\overline{S}$. Furthermore, the response durations $\tau_R$ obtained from the peak of the cross-projection magnitude temporal profiles $\overline{S}_t$ clearly capture the timing of meaningful structure that is visually apparent in the CCEP traces. For present use, we plot error bars around $\tau_R$ representing the limits where $\overline{S}_t$ exceeds 98% of $\overline{S}(\tau_R)$.

Synthetic response traces, shown in Fig 7 (and S1 Fig) give the reader a set of simple illustrations to develop intuition for $\overline{S}_t$. Notably, responses that are mirrored in voltage or mirrored in time produce the same peak response magnitude, at the same duration (Fig 7G–7L). Splitting a response in two parts and separating them in time does not change the peak cross-projection response magnitude (Fig 7A–7C). Addition of noise to a response will not change the response duration, and only decreases the cross-projection magnitude at very high levels of noise (Fig 7D–7F). Note that cross-projection magnitude profile $\overline{S}_t$ for a sustained fixed voltage offset increases by $\sqrt{time}$, as seen in Fig 7. This is a consequence of the fact that the measure is semi-normalized—one of the single-trial vectors in the dot product is normalized

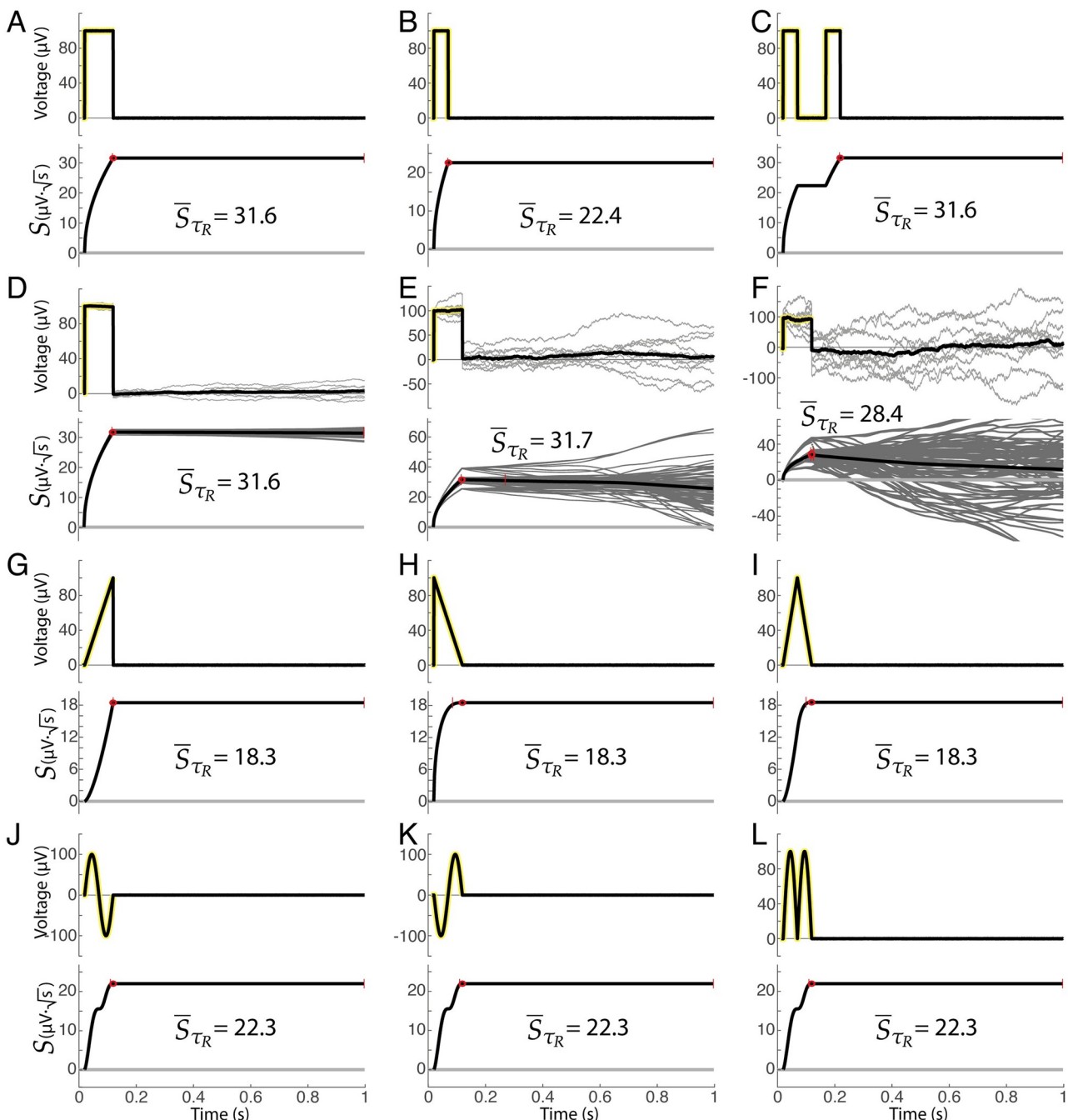

**Fig 7. Examples of projection magnitudes and profiles obtained with synthetic data. A.** A 100ms, 100$\mu$V synthetic square wave response (zero noise). **B**. 50ms/100$\mu$V square. **C**. Two 50ms/100$\mu$V square. **D**. 100ms/100$\mu$V square (low noise). **E**. 100ms/100$\mu$V square (intermediate noise). **F**. 100ms/100$\mu$V square (high noise). **G**. Ramp up to 100$\mu$V over 100ms (zero noise). **H**. Ramp down from 100$\mu$V over 100ms. **I**. Ramp up to 100$\mu$V over 50ms then down to 0$\mu$V over 50ms. **J**. Sinusoid (peak ±100$\mu$V) over 100ms. **K**. Inverted sinusoid. **L**. Absolute value of sinusoid.

$(\tilde{V}_k(t) = V_k(t)/|V_k(t)|)$, while the other is not. Because of this semi-normalization, the noiseless $\overline{S}_t$ profile examples in Fig 7 plateau rather than decrease when the signals return to zero after the synthetic feature—a result of the fact that there is no anti-correlation in pairwise comparisons to reduce the value of the sum in $\overline{S}_t$ from its peak.

How $V(t)$ is normalized prior to cross-projection has a marked effect on how significance is determined, as illustrated in Fig 8. The un-normalized approach ($V_k(t)V_l(t)$) is sub-optimal because trials with large amplitude are relatively over-emphasized, even when their shape does not reflect the most characteristic structure. Conversely, fully-normalized projections $\tilde{V}_k(t)\tilde{V}_l(t)$ are sub-optimal because they measure higher significance for shorter lengths of

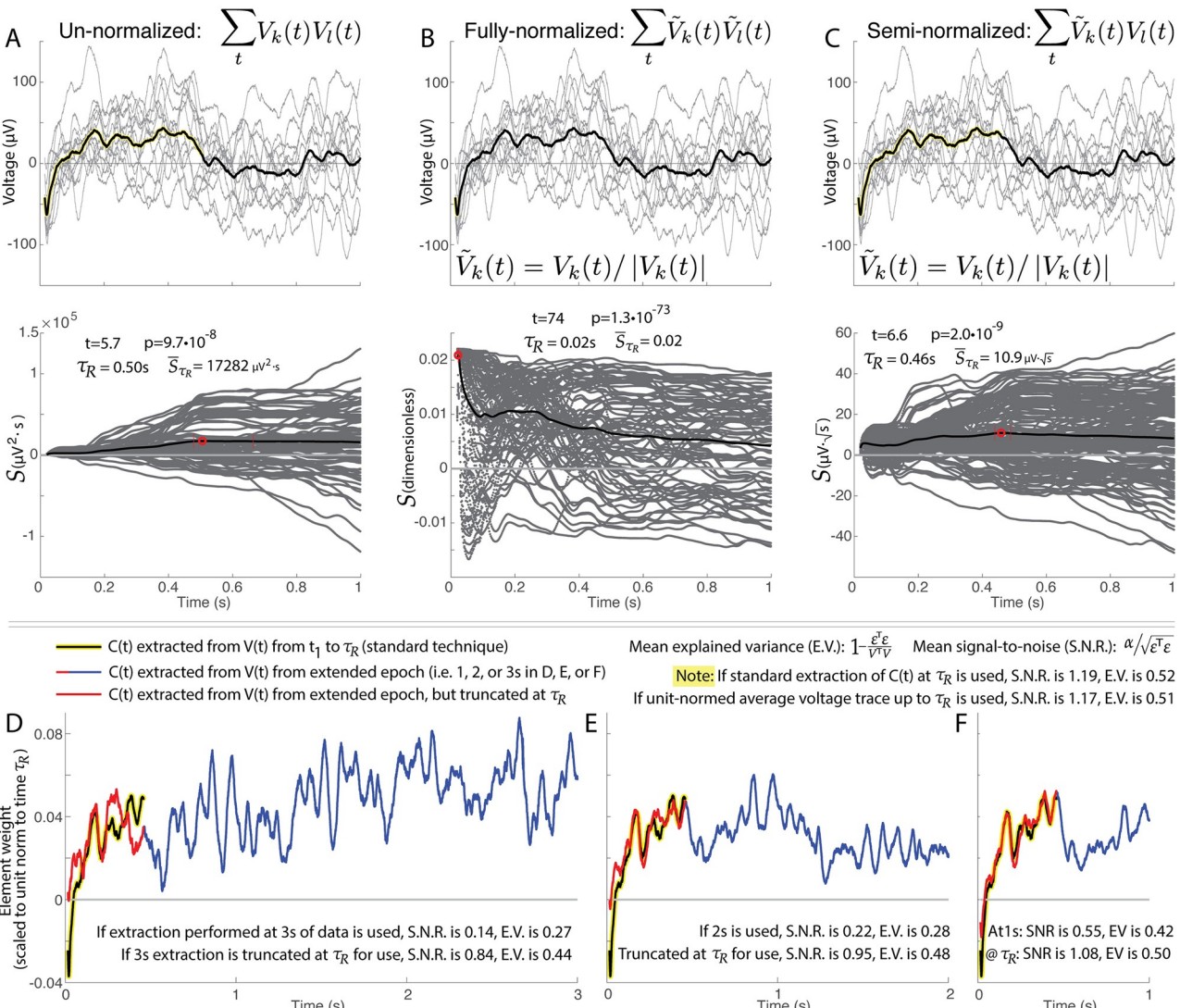

**Fig 8. Illustrations of different normalizations of single-trial cross projections.** As discussed in the manuscript, different trials $V_k(t)$ and $V_l(t)$ may be compared with each other directly, or after normalization with $\tilde{V}_k(t) = V_k(t)/|V_k(t)|$. **A**. Un-normalized projections $V_k(t)V_l(t)$ are sub-optimal because trials with large amplitude are over-emphasized in comparison with trials of lower amplitude but more characteristic structure. **B**. The time-resolved structure of fully-normalized projections $\tilde{V}_k(t)\tilde{V}_l(t)$ are sub-optimal because they dramatically favor early transients and cannot resolve temporally-sustained structure. **C**. Semi-normalized projections are optimal in that they balance emphasis of amplitude and sustained structure between trials. *Panels D-F show the same sample data as A-C, and illustrate the effect of extracting the canonical response from different epochs of time.* In the "standard" extraction approach we have illustrated so far, $C(t)$ is discovered using linear kernel PCA from **V(t)** over the isolated time interval from $t_1$ to $\tau_R$ (black line with yellow highlight). We can also unit normalize the average voltage $\overline{V(t)}$ over the $t_1$ to $\tau_R$ interval, though the explained variance and signal-to-noise are slightly worse. **D**. If a $C(t)$ is extracted using linear kernel PCA from $t_1$ to $t_2 = 3$ s (blue+red compound trace), the explained variance and signal-to-noise is very poor due to the introduction into the algorithmic process of a large amount of unnecessary noise from the time following $\tau_R$, even if the extracted form is truncated at $\tau_R$ for parameterization (red trace). **E and F**. As in (D), but for $t_2 = 2$ s (E) and $t_2 = 1$ s (F). Note how the shapes converge as $t_2$ decreases.

data (favoring early transients), and are unable to resolve sustained structure over time (as the normalization factor penalizes added datapoints). Semi-normalized projections, $\tilde{V}_k(t)V_l(t)$, nicely balance an emphasis between response amplitude and sustained structure.

Response duration $\tau_R$ captures the point in time where the signal produced from stimulation becomes indistinguishable from zero (Fig 3). Automated quantification of response duration, rather than visual identification, is important because there is wide variation in duration across pairs of stimulated-at and measured-from brain sites (i.e. Fig 1E). It is also very important because it enables further discovery and robust parameterization of structure in the data: by taking only the segment of data up to the response duration when performing parameterization, unnecessary noise that follows this time does not confuse or diminish the algorithmic process (as illustrated in Fig 8D–8F).

In principle, a response onset/beginning time, $\tau_B$, could be calculated moving backward from the discovered response duration, e.g. search through a profile of $\tau_B - \tau_R$ once $\tau_R$ has been discovered. For the present application, that is felt to be unproductive since conduction times between stimulated electrode pairs and measured responses is of the same order as the initial $\tau_B$ ($\sim$15ms). However, calculation of onset/beginning time would be useful in other, future, contexts, where there is a clear delay between the stimulation and response onset. For example, application of CRP and calculation of $\tau_B$ may be useful in the study of visual or auditory evoked responses, where we know that there is a lag between visual presentation and physiological response that can change in the context of disease (e.g. visual evoked potentials increasing in latency in the context of optic neuritis [25]).

## Parameterization of single trials by canonical CRP shapes, $C(t)$, magnitude of the voltage response, $\alpha'_k$, and the residual, $\varepsilon_k$

The discovery of response duration defines the information-rich epoch of data following stimulation, and allows for isolation of the characteristic induced response shape $C(t)$, by using linear kernel PCA on $\mathbf{V(t)}$ over the isolated time interval from $t_1$ to $\tau_R$. Our data-driven CRP approach is an important tool to move analysis of these brain stimulation data past the level of characterization by eye, discovering $C(t)$ empirically (rather than assuming a pre-defined shape). With CRP, researchers can identify and compare different response shapes across stimulation and recording brain sites in different patients using a unified quantification. The formalism is adopted from the field of functional data analysis [23, 24] and allows us to express single trials of the voltage response as $V_k(t) = \alpha_k C(t) + \varepsilon_k(t)$ (Fig 4). This representation allows single trials to be summarily characterized by normalized projection weight ($\alpha'$, in units voltage), signal-to-noise ratio, and explained variance (Fig 5). Importantly, the CRP technique is quite robust and performs well with diminishing signal in the presence of constant noise (which we have found explicitly, using synthetically-generated responses—illustrated in S1 Fig). Quantifying effect size and statistical significance in this way helps to compare many different response shapes (whether short or long) within one framework, and opens up the possibility to explore data in the hypothesis-preselected and divergent paradigms [10]. While our illustrations consist of a low number of trials (10–15), the technique works easily and is associated with higher statistical significance when a much higher number of trials are obtained (S2 Fig). Seemingly dissimilar responses may be statistically compared with one another without difficulty, as illustrated in Fig 5. Of note, the N1/N2 shape, when present, is clearly and effectively captured (e.g. second row of Fig 5).

While the numerical values of $\alpha$ are not intuitive, $\alpha'$ is normalized by the square root of the number of samples in $C(t)$ (i.e. in $t_1$ to $\tau_R$ interval) and roughly captures the average voltage deflection from zero during the significant response interval. $\alpha'$ is comparable to the root-

mean-squared metric that has been shown to be useful in this context [26], but is weighted only by the empirically-discovered significant interval of the response, rather than a pre-selected epoch defined by the researcher.

The different metrics to quantify and compare response size and significance will be most useful depending on the context. Normalized projection weight $\alpha'$ is useful to compare whether one stimulation-response pair has a larger average voltage. However, one brain site might have less baseline activity (as quantified by voltage) than some other site, due to a different cellular milieu or organization; in this case the magnitude of the voltage deflection compared to residual or the explained variance in the signal by the stimulation $\left(\text{i.e. } \alpha_k / \sqrt{\varepsilon_k^\mathsf{T} \varepsilon_k} \quad \text{or} \quad 1 - \frac{\varepsilon_k^\mathsf{T} \varepsilon_k}{V_k^\mathsf{T} V_k}\right)$ are more meaningful comparators.

Canonical shapes $C(t)$ can be compared across brain sites and patients by taking the dot products to compare similarity. As such comparisons mature, one might account for variations in anatomy and physiology that preserve overall response shape but not conduction speed by performing scaling in time (i.e. by "stretching" $C(t)$ with established techniques [27, 28]).

Note that the ability of $C(t)$ to capture the signal in, and explain the variance of, the voltage responses is diminished if one applies the PCA extraction on longer segments of data, or uses a unit-normalized version of the averaged trace (CCEP), as seen in Fig 8D–8F.

Although this manuscript has concentrated on exploring interactions between stimulated-at and measured-from brain sites, one need not measure at a different site than was stimulated to apply our methodology. One may stimulate and measure the evoked response at the same site, applying this parameterization to the measurement, but ensuring that the beginning time of analysis, $t_1$ is chosen to be well after the stimulation artifact and volume conduction effects have passed [29].

The residual term, $\varepsilon_k(t)$, is a signal that reflects all local brain activity not directly linked to stimulation timing, combined with measurement noise (i.e. from amplifiers and the environment). For example, if a researcher wishes to examine non-phase-locked oscillatory (rhythmic) activity resulting from the stimulation, they should calculate this from $\varepsilon_k(t)$ rather than $V_k(t)$, since the shape of the deflections of the evoked potential $C(t)$ will have corresponding power in the Fourier domain. For example, a positive deflection in a component of $C(t)$ lasting 100ms will have power at $1/(2 \cdot 0.1s) = 5$Hz, but not be an oscillation. Extracts of broadband spectral activity (spread across all frequencies according to a power-law form, but often captured at high frequency by researchers) that capture local brain activity [30] might be best extracted from $\varepsilon_k(t)$ rather than $V_k(t)$.

The $\varepsilon_k(t)$ term can be used as a tool to understand changes in the shape of the response after external conditions have been applied to perturb brain state. After performing a set of stimulations and parameterizing the responses, one might administer a pharmacologic agent, perform a behavioral analysis, apply therapeutic stimulation, or observe a global state change (e.g. transition from waking to sleep, etc). Stimulations may then be re-performed with the brain in the perturbed state, but the responses are parameterized according to the *original* $C(t)$, obtaining a new set of residuals $\varepsilon_k^n(t)$. Then, the extraction and parameterization described in this manuscript is applied to $\varepsilon_k^n(t)$ rather than $V(t)$: If any significance is identified, then the resulting *new* $C^n(t)$ that emerges reveals the structure introduced by the perturbation to brain function (pharmacologic, stimulation therapy, awake/asleep, etc).

## Characterizing significance, anomalous trials, and artifact

When considering whether a set of $N$ trials have a significant response to stimulation, the *extraction significance* defined above reveals how robust the shape is, providing a distribution

of $N^2 - N$ cross-projection magnitudes that may be tested versus zero for significance (e.g. 90 datapoints for a 10 trial set). It should be noted that there is some relationship between conjugate cross-projection magnitudes $\sum_t \tilde{V}_k(t) V_l(t)$ and $\sum_t \tilde{V}_l(t) V_k(t)$, but they are not the same (as seen clearly by the difference between red and green datapoints in Fig 9D. This relationship is addressed by a balanced downsampling so that trial-pairs of projections are only included once (as described above). The distribution of p-values obtained from many iterations of null data are evenly distributed on the 0-to-1 interval, indicating that the extracted significances are statistically appropriate (S3 Fig).

As an alternative to the *extraction significance*, one may calculate the *parameterization significance* based upon the single trial parameters noted in Table 1, of which there are $N$ datapoints for each parameter (e.g. 10 datapoints for a 10 trial set). For test of significant response (versus no response) $\alpha_k$ is the most useful, because it would be expected to be distributed around zero (insignificant) for spurious discovered structure $C(t)$.

When comparing different stimulation-response sets that have very different shapes, it can be quite useful to compare the distributions of parameters between the two. One might say that a response is "significantly larger" than another by comparing one distribution of $\alpha'$ to another or a response is "more robust" by comparing $1 - \frac{\varepsilon_k^\mathsf{T} \varepsilon_k}{V_k^\mathsf{T} V_k}$ distributions (i.e. comparing their explained variance).

We may use the tools of this extraction and parameterization to characterize single trials within a set of stimulation-response measurements—anomalous single trials can be identified by individual comparison of the distribution of cross-projection magnitudes involving one trial to the all of the cross-projection magnitudes involving other trials. Trials with larger-

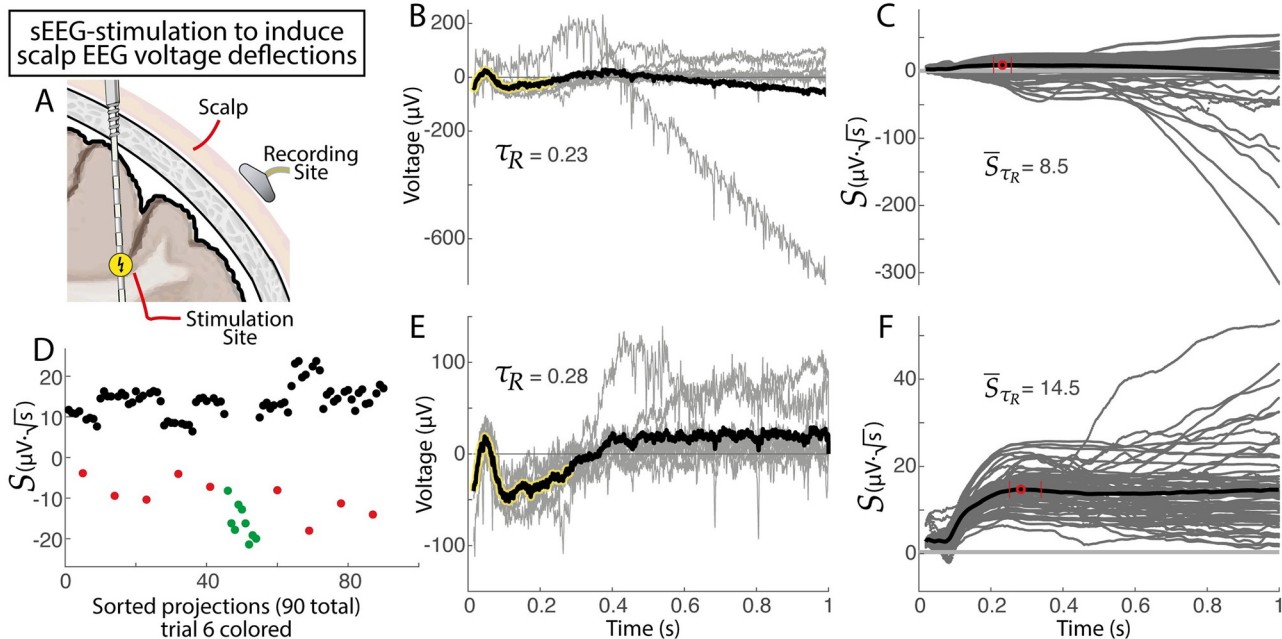

**Fig 9. Voltage deflections in the scalp EEG from intracranial sEEG electrical stimulation pulses, and automated artifactual trial identification. A**. Schematic, showing sEEG stimulation and EEG recording. **B**. Ten single-pulse EEG trials (gray) and average trace across trials (black). Note the clearly artifactual trial. **C**. Time-resolved projection magnitudes for trials from (B). **D**. Projection magnitudes at $\tau_R = 0.23$s, suggesting that trial #6 is artifactual (p = $1.8 \cdot 10^{-6}$, unpaired t-test comparing red+green vs black). Green dots indicate projections of normalized trial #6 into other trials, and red dots indicate normalized projections of other trials into trial #6. **E and F**. As in (B and C), with trial #6 removed. Note the change in $\tau_R$ from 0.23 to 0.28s and $\overline{S}_{\tau_R}$ from 8.5 to 14.5$\mu$V·$\sqrt{s}$.

than-average cross-projections may be interpreted as the "most representative" of the shape of the response. Conversely, as illustrated in Fig 9, trials with cross-projections that are far below the others can reveal artifactual trials for automatic rejection from the dataset (for automated artifact rejection, one can use projections at $\tau_R$ or at full duration $T$ sent for analysis). Note that there is a large and rich literature for identifying anomalous data [31], and it will be an interesting future direction to explore more comprehensively with data and methodology of this type.

## Biological interpretation of sEEG CCEP

Fig 1 shows that there are a wide variety of stimulation response shapes in the sEEG stimulation-evoked-potential responses, and the durations of these responses may be less than 100ms or last up to 600–700ms. This would be expected from brain surface ECoG measurements, where very different response shapes and durations could be evoked at the same measurement site depending on what site was stimulated [10]. One might hypothesize that the systematic study of waveform shapes $C(t)$ may provide understanding about the biology underlying these responses:

Could each $C(t)$ morphology type reflect a set of projections to different aspects of laminar architecture (e.g. different cell classes, or unique synaptic subtypes on pyramidal neurons) [32]? For example, in primary visual neocortex, differences in laminar pattern separate feed-forward and feedback connections across the 6 layers, allowing for the characterization of a visual hierarchy [33]. Feedforward connections preferentially terminate in the middle layer (layer 4), feedback connections preferentially avoid layer 4, while lateral connections terminate in roughly equal density across all layers. A different primary brain area, the motor neocortex, completely lacks a distinct layer 4. The hippocampus, which is archicortex rather than neocortex, only has 3 layers. One would therefore expect that a homologous input type, even if originating from the same source, would produce very different $C(t)$ in visual, motor, and hippocampal cortex. Therefore, consistent differences in $C(t)$ across these regions could inform new models of how intralaminar dynamics generate characteristic voltage responses.

Will future work find that specific shapes of $C(t)$ imply specific biology, such as pro-dromic versus anti-dromic propagation, long-track versus u-fiber white matter transmission, intracortical excitation (via axons that project laterally and remain within the gray matter), or thalamo-cortical relays [34]? We know for example that, for the divergent paradigm, evoked potentials may arrive with smoothly varying latency, duration, or polarity along adjacent sites in an sEEG lead traversing a natural axis in a brain structure (e.g. the body of the hippocampus in response to stimulation [18]).

The amplitude, width, and overall shape of voltage deflections are influenced by factors relating to the synchronous electrical activity produced in these neuronal populations. At the microelectrode scale, local field potentials have been shown to predominantly reflect coordinated synaptic inputs [35, 36]. For example, negative deflections in LFP recorded at the cortical surface can often represent current sinks generated by synchronized excitatory postsynaptic potentials (EPSPs) at apical dendrites of superficial pyramidal cells. In contrast, EPSPs at deeper cortical layers result in positive deflections in the same surface LFP. The width of an LFP deflection may therefore reflect the coherence of synaptic inputs, or may reflect the time-scale of charge influx, which is specific to the neurotransmitter type, signal transduction cascade, and channel dynamics that characterize each synapse [37].

## Applications in other scientific and medical contexts

Although we have illustrated this parameterization to the case of single-pulse electrical stimulation through sEEG electrodes, the approach might be applied in many other settings where

one wants to characterize a reliable response structure of unknown duration. A few such possibilities are:

- Evoked electrical and magnetic changes in the brain in response to visual or auditory stimuli are typically called event-related potentials (ERPs) (cf. [38, 39]). ERPs have been studied exhaustively in EEG, ECoG, and MEG, to study sensation, perception, cognition, memory, and other aspects of brain function [38, 40–44]. Event-related potentials have been studied to understand injuries and diseases of the brain and spinal cord [45–47], and are also used intraoperatively to dynamically to study the function of the spinal cord and brainstem (e.g. somatosensory evoked potentials—SSEPs [48] and brainstem evoked auditory responses—BAERs [49]). Much as in the case of the N1/N2 formulation described above, these ERP data typically focus on identification of a feature by the voltage at hand-picked latencies after the stimulus. The CRP approach detailed here would automate and simplify identification of structure and relative significance in the ERP. For the example case of ERPs in EEG, it is often said anecdotally that single trial signal is very low compared to the residual "noise"—application of CRP would allow one to quantify this explicitly.

- Early work with a similar formulation has been also useful for colleagues in neuroscience examining the EEG response to deep brain stimulation [50]. Our specific extraction and parameterization may fit nicely into their work, expanding on it by allowing for identification of the salient duration $\tau_R$ and single-trial parameterizations noted in Table 1.

- The hemodynamic response functions (HRFs) measured with fMRI have different shapes across different regions and laminae (cf. [51, 52]), and CRP might simplify the comparison of these in different voxels.

- A parameterization could be performed by replacing stimulation times with "discovered events" in ongoing brain data may be useful in examining electrophysiology studies such as action potential characterization and sorting in high-impedance microelectrode recordings.

- Brain state under anesthesia can affect CCEP shape [11]. One might apply CRP to a set of stimulations performed under one state of anesthesia, and apply the initial parameterization to a new set of stimulations performed during a different subsequent anesthesia state. Changes in $\alpha'$ or repeated CRP applied to the residual $\varepsilon(t)$ of this subsequent parameterization would reveal change in response structure that accompanies change in anesthesia.

- Somatosensory evoked potentials (SSEPs) are measured from the brain or spinal cord in response to electrical stimulation of the peripheral nervous system for medical diagnostics in the operating room and the clinic. In the operating room, these are a realtime diagnostic electrophysiology that can dynamically reveal impending injury so the surgeon can stop an action before causing permanent injury. In the clinic and hospital setting, these can be used to diagnose brain function in coma (diminished level of consciousness) after anoxia or traumatic brain injury [46, 53, 54]. Parameterization would dramatically simplify the nuance required by the electrophysiological technician who assists in these surgical procedures.

- Single-pulse electrical stimulation of the white and gray matter has been used for intraoperative connectivity mapping during surgery for tumor and epileptic focus resection [55, 56]. The utility of these diagnostics is still being explored [57], and CRP could help to simplify and standardize the interpretation of the CCEPs (the shape of which, as shown here, will vary dramatically), helping to identify the optimal approach for assistance during resection.

- Our ongoing work—as well as those of many colleagues [58]—is focused on the exploration of epileptic networks. With the advent of stimulation devices that can record and perform

open- or closed-loop stimulation [59–61], and can stimulate through 4 leads [62], brain stimulation for epilepsy is rapidly evolving. As we learn to stimulate networks in tandem at different cortical and thalamic sites, the ability to quantify connectivity during sEEG implants will help to drive better DBS and RNS system implantations [63].

## Limitations, alternative considerations, and future technical strategies

There are important limitations to consider when implementing CRP. By construction, this method cannot parameterize the timing of particular features. An example of this is the case where one wishes to quantify the propagation time between two areas (i.e. the latency). This is typically done by finding the first extrema (peak or trough) in the averaged response as an important time [6, 17, 64]. While this could be performed on $C(t)$, it is not explicitly built into the parameterization process. A change in output can have forms that are not easily tracked by the CRP technique, such as perturbations in the overall brain state affecting the amplitude of a specific deflection within the CCEP (while sparing other deflections) in subsequent stimulation trials; alternatively, timescales can change and the duration may get longer. CRP may not be useful in those settings (though the change would be quantified in correlated structure across the residual $\varepsilon(t)$).

By taking the peak magnitude of $\overline{S}_t$ as the duration, a late resurgence of structure ("blip") following a period of relative insignificance will only contribute if it can overcome any intervening loss in cross-correlation to create a higher peak in the projection profile. Notably, we have not yet observed this in our studies, but it remains an important possibility to be aware of.

The reader should be aware of two frequent artifactual conditions, illustrated in Fig 6. In the first of these (Fig 6C and 6D), a seemingly insignificant response has a very significant brief structure at the beginning of the examined period. This situation may arise when a latent effect of stimulation artifact "carries forward" into the window being considered. Determining what is stimulation artifact and what is brief evoked neural activity is a nuanced topic that we defer to future study. The second artifactual condition to consider (Fig 6E and 6F) is the case where a set of responses appear to be complete noise, but the time-resolved projection magnitude $\overline{S}_t$ grows steadily in time. Inappropriate baselining of the data produces this—we also defer comprehensive exploration of this to future treatment.

As opposed to quantifying time-domain (i.e. raw voltage) changes, one might instead study responses to single pulse electrical stimulation in the frequency domain. Broadband changes in the frequency domain are of particular appeal since their shapes may be interpreted generically as increases in firing rate [30, 65–67], without the need to interpret polyphasic shapes as we do in this manuscript. Crowther, et. al. and Kundu et. al. [13, 68] showed that broadband changes can effectively identify interactions between brain areas, and it is very likely that broadband changes and raw voltage changes have complementary information, which has been shown for ECoG responses to visual stimuli [41]. Frequency-domain changes that are peaked at a particular frequency (rather than broadband) can reveal stimulation-evoked brain rhythms (oscillations), and are a topic of future study. As noted above, one must be careful when inferring the presence of a rhythm purely from examining responses in the frequency domain, since a simple voltage deflection (like many of those seen in Fig 1), will have power at a frequency that is the inverse of the width of the voltage deflection.

Future exploration might expand this parameterization approach in a number of different directions:

- $C(t)$ could be chosen in different manners than we have, such as: using the simple average trace (i.e. $C(t) \rightarrow \overline{V(t)}$—we have anecdotally found that using the simple mean as the

canonical form, rather than the PCA-based extraction, increases $\alpha'$, while reducing the SNR, as seen in S4 Fig); using the "most representative" single trial, identified by the trial that has the largest average cross-projection when compared with other trials (e.g. the first trial, indicated by blue dots, in Fig 6E and 6F), truncated to $\tau_R$; a globally-defined average shape, such as one chosen from the average in a corresponding brain site over many patients [14]; or canonically-defined shape, like the "N1/N2" shape.

- As noted above, a beginning time $\tau_B$ could be calculated by moving backward from the optimal duration and recalculating the $\overline{S}_t$ profile, but over $\tau_B - \tau_R$ once $\tau_R$ is known.

- It is possible to calculate the "significance of the leftover", by performing projections on the leftover matrix $V_k(\tau_R - to - T)$, and defining the new $\tau_R$ as the value for which "leftover" cross-projection magnitude drops below a particular significance level. There are problems with this, because choice of $T$ is somewhat arbitrary without further constraint, so the threshold will also be arbitrary.

- In our present application, error bars (uncertainty) around $\tau_R$ represent the limits where $\overline{S}_t$ exceeds 98% of $\overline{S}(\tau_R)$ (since $\overline{S}_t$ reflects a distribution). Future approaches might employ a more nuanced approach to quantify this.

- There is very little jitter in the response onsets of these CCEPs. In other contexts, such as ERP research, there is known variation in response onset, and expanded approaches will be needed in order to align trials temporally prior to parameterization.

- It is possible to calculate cross-projections using an alternative approach: Instead of normalizing one trial by its norm, and not normalizing the other, one can normalize both trials by the square root of their respective norms. This has the effect of tightening the distribution of projections—generating higher extraction significance, but also de-emphasizing anomalous trials (i.e. those that are most representative, or those that are most likely to be artifactual).

- It may be useful in future studies to keep additional columns of $\mathbf{X}$ in the linear kernel PCA to study variation across trials (i.e. the second-order moment in the data), rather than the first column alone which is our canonical CCEP shape $C(t)$ (a robust approximation of the mean).

- Future treatments might examine the effect of temporal dilation in a response—where the shape of the physiologic response is prolonged or contracted due to disease, medication, etc. The field of functional data analysis has developed "time warping" approaches [27, 28] for just this purpose, and they can be applied directly to the CRP parameterization.

- It will be interesting to explore an optimal CRP parametrization for multimodal brain data (e.g. [69–71]). Here the parametrization may further reflect cross-modal spatial and temporal dependencies.

## Supporting information

**S1 Fig. Artificially-generated evoked responses with variation in signal to noise. A**. An artificial signal trace, normalized to variance of 1. **B**. 10 trials of brown-noise (i.e. random walk) timeseries, with each normalized to z-score of 1. Brown noise generated by cumulative sum of random data on -0.5 to 0.5 interval and subtracting off of running mean. **C**. Response duration (left) and timecourse of projection weights (right) extracted from synthetic traces with noise traces from (B) added to signal trace at ratio of 3-to-1. **D**. Response duration (left) and timecourse of projection weights (right) extracted from synthetic traces with noise traces from (B)

added to signal trace at "mild" variable ratios of 1.2 to 3.0 in 0.2 intervals. **E**. Response duration (left) and timecourse of projection weights (right) extracted from synthetic traces with noise traces from (B) added to signal trace at "extreme" variable ratios of {0.38; 0.47; 0.59; 0.74; 0.94; 1.18; 1.48; 1.86; 2.34; 2.94}. **F**. Parameterization of the artificial evoked responses of constant signal-to-noise ratio of 3-to-1 (from (C)). **G**. Parameterization of the artificial evoked responses of the "mild" variable signal-to-noise ratios of 1.2 to 3.0 in 0.2 intervals (from (D)). **H**. Parameterization of the artificial evoked responses of the "extreme" variable ratios of {0.38; 0.47; 0.59; 0.74; 0.94; 1.18; 1.48; 1.86; 2.34; 2.94} (from (E)). **I**. Extracted C(t) for different noise levels overlaid on top of original artificial form. **J**. Single-trial noise residuals for different noise levels. **K**. Single-trial $\alpha'$ to noise residual (SNR) for different noise levels. **L**. Single-trial explained variance for different noise levels. **M**. Single-trial ratio of coefficient $\alpha'$ to input SNR for different noise levels. Differences between extreme variable traces (yellow) in (J) and (M) are due to shorter C(t). This shorter C(t) may be related to added correlated deviation toward zero by the brown noise statistics disproportionally contributing at higher noise levels. (TIF)

**S2 Fig. Example of parameterization with a large number of trials.** Measurement is from a dorsal insular contact in response to stimulation of white matter in the orbitofrontal cortex. **A**. Stimulation was performed 69 times. Artifact rejection was at a threshold of $p < 10^{-10}$, resulting in rejection of 6 trials. The extraction was robust with an associated t-value of 149. **B**. The first 10 trials from (A) were parameterized in an identical fashion. No trials were identified as artifactual, and the associated t-value for extraction was 23. Note that the response duration ($\tau_R$), mean projection at response duration ($\overline{S}$), and scaling coefficient ($\alpha'$) were all nearly identical. However, the statistics of the parametrization were much more robust for 69 trials. The averaged explained variance and SNR were slightly higher for 10 trials (as might be expected). (TIF)

**S3 Fig. Generation of many artificially-generated sets of pure-noise data to validate statistics.** **A**. Top: An example of a single brown-noise (i.e. random walk) timecourse. Bottom: A 10-trial set of brown noise timecourses. **B**. A histogram of extraction significances from 20,000 surrogate sets of brown-noise timecourses. **C**. Top: An example of a single white-noise timecourse. Bottom: A 10-trial set of white-noise timecourses. **D**. A histogram of extraction significances from 20,000 surrogate sets of white-noise timecourses. Because histograms of p-values show a flat distribution over the 0-to-1 interval in (B) and (D), we may infer the statistical method is well calibrated for null models. (TIF)

**S4 Fig. An illustration of calculating the canonical shape from linear kernel PCA or from the simple mean.** (A-C) are from the example in the middle row of Fig 5 and (D-F) are from Fig 8. **A**. The averaged voltage response, is shown with a black line, and the significant portion of the response is highlighted (i.e. up to $\tau_R$). **B**. C(t) calculated from linear kernel PCA (blue) and from the simple mean (red). **C**. Parameterizations calculated from linear PCA vs mean voltage extractions. **D-F**. As in (A-C), for the example from Fig 8. (TIF)

## Acknowledgments

We are grateful to the patients who volunteered their time to participate in this research and to the staff at St. Marys hospital. Oliver Unke provided valuable comments on the manuscript.

We would also like to thank a reviewer from the manuscript "Basis profile curve identification to understand electrical stimulation effects in human brain networks" who suggested inquiry along these lines, the exploration of which lead to this present work.

## Author Contributions

**Conceptualization:** Kai J. Miller, Klaus-Robert Müller, Dora Hermes.

**Data curation:** Gabriela Ojeda Valencia, Dora Hermes.

**Formal analysis:** Kai J. Miller.

**Funding acquisition:** Kai J. Miller, Klaus-Robert Müller, Gregory A. Worrell, Dora Hermes.

**Investigation:** Kai J. Miller, Gabriela Ojeda Valencia, Harvey Huang, Nicholas M. Gregg, Dora Hermes.

**Methodology:** Kai J. Miller, Klaus-Robert Müller, Dora Hermes.

**Project administration:** Kai J. Miller, Dora Hermes.

**Resources:** Kai J. Miller, Gregory A. Worrell, Dora Hermes.

**Software:** Kai J. Miller.

**Validation:** Harvey Huang.

**Visualization:** Kai J. Miller.

**Writing – original draft:** Kai J. Miller.

**Writing – review & editing:** Kai J. Miller, Klaus-Robert Müller, Gabriela Ojeda Valencia, Harvey Huang, Nicholas M. Gregg, Gregory A. Worrell, Dora Hermes.

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
