## [Decision Letter · Decision Letter 0]

10 Nov 2022

Dear Dr Miller,

Thank you very much for submitting your manuscript "Canonical Response Parameterization: Quantifying the structure of responses to single-pulse intracranial electrical brain stimulation" for consideration at PLOS Computational Biology.

As with all papers reviewed by the journal, your manuscript was reviewed by members of the editorial board and by several independent reviewers. In light of the reviews (below this email), we would like to invite the resubmission of a significantly-revised version that takes into account the reviewers' comments.

While both reviewers agreed that the approach is novel and valuable, they have raised concerns regarding 1) quantification of algorithm performance, in particular critical factors influencing performance including the number of trials 2) validation and clarification regarding the domain of application

We cannot make any decision about publication until we have seen the revised manuscript and your response to the reviewers' comments. Your revised manuscript is also likely to be sent to reviewers for further evaluation.

Sincerely,

Hayriye Cagnan

Academic Editor

PLOS Computational Biology

Lyle Graham

Section Editor

PLOS Computational Biology

While both reviewers agreed that the approach is novel and valuable, they have raised concerns regarding 1) quantification of algorithm performance, in particular critical factors influencing performance including the number of trials 2) validation and clarification regarding the domain of application

Reviewer's Responses to Questions

**Comments to the Authors:**

Reviewer #1: The authors introduced a new method for quantifying the structure of the response to SPES. This method does not need a predetermined waveforms such as N1 or N2 by the examiner. Instead it employs semi-normalized dot-product projection to extract non-artifactual true response, determine the response duration, and then extract a characteristic shape for each recording contact using the Canonical Response Parameterization (CRP). They applied this methodology to the 2 patients with SEEG investigation, who underwent CCEP recording (10 stimulation per stimulus site). The concept is interesting in that we do not need a priori waveform as the previous SPES/CCEP literatures have defined, and that the method produces the canonical waveform and gets rid of the residual activities. The attempt is novel and interesting, but this new method needs some clarification for publication in the PLOS Computational Biology.

#1 Impact on artifact reduction

It is interesting to note that this method can find the outlier waveform (out of 10 waveforms obtained by 10 trials of stimulation). This is of clinical use when only the number of stimulation trials is small (10) as in this study. This may not be so effective for the stimulation trials over 50 trials. It would be nice if you have actual CCEP dataset of >50 trials and compare the effect of this method upon artifact reduction (or do some sort of simulation).

It is also interesting to know how this method can get rid of the epileptic activity (usually larger than CCEP) that overrode (superimpose) on the CCEP response (which may occur only in some trials).

#2 Significance of the method for clinical evaluation

As the authors pointed out typical N1/N2 waveforms are not always available especially in the case of SEEG recording. It would be nice to show more examples (CCEP findings form all the electrode contact (grey matter) for each stimulus site, and if possible, CCEP findings for more stimulus sites), since only a part of the CCEP waveforms is shown. It would be nice to see the effect of this method on the adjacent large CCEP response as well as the remote, relatively small CCEP responses. In addition, it is recommended to show these results together with their anatomical locations so that readers can compare the findings with the previous literatures.

From clinical point of view, if the authors can compare 1) mere averaging and 2) the results of their method (showing only C(t)) by showing their waveforms, then the readers from the clinical field can understand the utility of this method.

#3 Limitations

While the authors do acknowledge the timing issue of the CRP in limitations, the authors did not mention the small number of patients they studied (N = 2). The authors had better mention this since the application in the larger patient group may reveal some other important features of this new methodology.

Reviewer #2: Miller et al. present a new method for data-driven characterization of single-trial electrophysiological responses to neural stimulation. This method is developed in the context of “cortico-cortical evoked potentials” (CCEPs) in the human brain. These CCEPs are shifts in the (usually bulk) electrical field at site X induced by (usually bulk) electrical stimulation at site Y. In order to understand how electrical drive at one site (X) affects the response at another site (Y), a conventional approach has been to characterize CCEPs (evoked at site Y) via the timing and of their “peaks”, especially the so-called N1/N2 maxima of the mean stimulation-evoked potential, which occur around 100 and 200ms post-stimulus. However, the authors argue that, for many pairs of cortical sites, the CCEPs do not exhibit (or are not well-characterized by) the N1/N2 components, so that a more general and flexible technique is required. The authors further argue that their approach (via linear kernel PCA) also allows the characterization of the duration of the evoked response, the computation of explained variance, as well as single trial response “magnitudes” and other features of the neural response.

This work is well illustrated and well argued, and it addresses a cornerstone methodological need for answering empirical questions around brain stimulation. This is valuable science, which certainly will make a contribution to the literature and to analytic practices of electrophysiological data.

The main drawback of this manuscript is that, although this method is clearly valuable and an advance on standard practice in human CCEP work, the manuscript does not provide the reader with (i) sufficient information about the conditions under which the method will work best and worst; and (ii) sufficiently detailed validation of the basic approach and statistical analysis methods. To be clear: my concern here is not that the method is invalid, but rather that its limits and domain of application are not clearly specified; this valuable work will be more immediately useful to other scientists if the limitations and assumptions are made clearer.

### Main Points ###

* 1 * There appear to be several assumptions made in applying the linear kernel PCA method and using it to characterize the single-trial parameters and associates statistics. Please provide the reader with more information about these, and their justification. For example, does the model make assumptions about (i) the consistency of the timing of peaks and troughs across trials and (ii) the consistency of the magnitude of the signal waveform across trials?

Related to consistency of timing of peaks across trials: If the true underlying signal is a sharply ramping function (e.g. ramping from -1 to 1 in 20 milliseconds), will this not appear as a temporally “blurred” slowly ramping component (e.g. ramping from -1 to 1 in 80 milliseconds) if there is 50 ms of timing jitter from trial to trial? If the peak of a CCEP became later and later across stimulations (e.g. due to some kind of circuit adaptation effects), how would this affect the extracted C(t) curve.

Related to consistency of magnitude over trials: suppose that the neural circuits exhibit a form of adaptation in its response to stimulations, so that responses decay in magnitude over trials (i.e. the noise retains constant variance but the signal variance decreases over trials, while the signal waveform maintains the same shape). How would this affects the performance of the linear kernel PCA method?

* 2 * Are the statistical methods well-calibrated? When data are generated from a null model, do they indeed produce 5% false positive rates in simulated data, when the alpha value is set to 5%? Does the statistical approach control the false-positive rate (nominally 5%) at the level of each individual time point within a trial, or does the statistical approach control the false positive rate at the level of the whole trial, or the whole dataset?

When testing the calibration of the statistics, it is important to be clear about the assumptions about the nature of the noise on each trial. Are there assumptions about (i) the distributional form of the noise, (ii) the independence of noise samples across time-points and (iii) the consistency of noise-variance across trials? Generally, we expect that the performance of the method may be sensitive to violations of some of these assumptions, and robust to violations of others, but which are which? It should be possible to explore some of these points in simulation, as in Figure 7.

* 3 * Regarding the method for determining the duration of the response, the tests in Figure 7A-C are a simple and beauitful self-consistency demonstration for the proposed method. However, does this self-consistency effect persist in the presence of noise? If not, it would be helpful to give the reader some sense of which kinds of noise levels are most likely to lead the method to lose this self-consistency. Is there a way for the end-user to know what level of noise they are dealing with in empirical data?

* 4 * When normalizing the single-trial cross-projections (Figure 8), the authors emphasize that a semi-normalized approach is best. In this approach, we compute the dot product between one unit-normalized vector and another unnormalized vector. I wonder if there is an alternative that is superior: instead of normalizing by the length of the V_k vector (|V_k|), normalize by the geometric mean of the lengths of the V_k vector and the V_l vector (sqrt(|V_k||V_l|)? The advantage of this approach, as far as I can tell, is that it would give less weight to outlier trials, and would be less susceptible to the influence of individual low-amplitude trials, because it combines information from both the row and the column in the P(k,l) matrix.

* 5 * Line 467: “ In the first of these (Figs. 6C and D), a seemingly insignificant response has a very significant brief structure at the beginning of the examined period. This situation may arise when a latent effect of stimulation artifact “carries forward” into the window being considered. Determining what is stimulation artifact and what is brief evoked neural activity is a nuanced topic that we defer to future study.”

Since cortico-cortical axonal conduction latencies are typically 10ms or longer, why not exclude (by default) the first 10ms post-stimulation from the process used to extract C(t) and measure the response duration? This may not fully address the issue, but it should reduce its influence?

* 6 * Line 472: “The second artifactual condition to consider (Figs. 6E and F) is the case where a set of responses appear to be complete noise, but the time-resolved projection magnitude S ¯t grows steadily in time. Inappropriate baselining of the data produces this - we also defer comprehensive 475 exploration of this to future treatment.”

I understand that there is not space for a full treatment of this point, but it seems important to add a couple of sentences here, elaborating on how inappropriate baselining can be avoided and / or how the end-user can detect when the baseline is incorrect.

Furthermore, it seems important to emphasize the importance of baselining for the success of this method. In many dimensionality reduction methods, the results are invariant to the centering approach (i.e. you get the same results if you z-score the data and if you don’t), but with this method, it appears that the absolute magnitude and sign of the voltage time-courses are being (successfully) exploited to determine which trials are more important and contain more signal. Maybe it would be worth emphasizing, around line 93, that the data are not zero-centered on each trial, but they should instead be centered relative to an independently-measured baseline reference.

* 7 * The Introduction of the manuscript would be more compelling if it stated, earlier on, the scientific gap or need that is filled by the present work. It would be nice if the manuscript stated very early on, both: (i) the shortcomings of current approaches, and (ii) the top reason(s) why this new method is important and useful. There is plenty of text making these points in the manuscript, but this motivational info is spread across the Intro, Results, and Discussion.

### Typos / Minor Points ###

** The method proposed here assumes that there is a single consistent response (the C(t)) which we wish to extract. Are there conditions under which it would make sense to extract multiple responses, i.e. additional columns of the matrix X, line 139, and would these additional responses need to be orthogonal?)

Line 18: “To address this, we recently introduced a set of four basic paradigms for interpreting CCEP data” — I think that it would be more accurate to say that terminology was recently introduced, or a conceptual framework was introduced, as these paradigms are many years old?

Line 36: “stimulated-at” can be “stimulated”

Line 107: “Because S^bar may be thought of as a measure of mutual information between responses…” — Please justify or elaborate on this point.

Tiny p-values: In some place, e.g. in the caption of Figure 5, the manuscript reports p-values of the form p << 10^{-34}, but such numbers are so vast (or tiny) that they are almost meaningless. I am not sure of the best way to handle this, but maybe something like “p approximately 0”, or “p-value negligible” or “p << 0.001” is better?

Figure 6A: What is “extraction significance” in the title?

Line 365: “How the recorded C(t) curves isolated in these regions of the brain differ from one another, if similar input types/regions can be approximated, might serve as an informative model to understand how intralaminar dynamics result in characteristic voltage responses.”

… Perhaps this would read better as: “Therefore, consistent differences in C(t) across these regions could inform new models of how intralaminar dynamics generate characteristic voltage responses.”

**Have the authors made all data and (if applicable) computational code underlying the findings in their manuscript fully available?**

Reviewer #1: Yes

Reviewer #2: Yes

PLOS authors have the option to publish the peer review history of their article (what does this mean?). If published, this will include your full peer review and any attached files.

Reviewer #1: No

Reviewer #2: No
---

## [Decision Letter · Decision Letter 1]

14 Apr 2023

Dear Prof Miller,

We are pleased to inform you that your manuscript 'Canonical Response Parameterization: Quantifying the structure of responses to single-pulse intracranial electrical brain stimulation' has been provisionally accepted for publication in PLOS Computational Biology.

Best regards,

Hayriye Cagnan

Academic Editor

PLOS Computational Biology

Lyle Graham

Section Editor

PLOS Computational Biology

Reviewer's Responses to Questions

**Comments to the Authors:**

Reviewer #1: The author generally responded to the concerns raised by the reviewer. I hope this method is of use for those who would like to evaluate CCEPs in the SEEG circumstances.

Reviewer #2: The authors have addressed all of my concerns. I congratulate them on developing this impactful contribution to science and to neurological practice.

**Have the authors made all data and (if applicable) computational code underlying the findings in their manuscript fully available?**

Reviewer #1: Yes

Reviewer #2: Yes

PLOS authors have the option to publish the peer review history of their article (what does this mean?). If published, this will include your full peer review and any attached files.

Reviewer #1: No

Reviewer #2: No

---

## [Editor Report · Acceptance letter]

22 May 2023

PCOMPBIOL-D-22-01199R1 

Canonical Response Parameterization: Quantifying the structure of responses to single-pulse intracranial electrical brain stimulation

Dear Dr Miller,

I am pleased to inform you that your manuscript has been formally accepted for publication in PLOS Computational Biology. Your manuscript is now with our production department and you will be notified of the publication date in due course.

With kind regards,

Livia Horvath
